# Programed Thermoresponsive Polymers with Cleavage-Induced Phase Transition

**DOI:** 10.3390/molecules27186082

**Published:** 2022-09-18

**Authors:** Yukiya Kitayama, Yasumichi Yazaki, Junya Emoto, Eiji Yuba, Atsushi Harada

**Affiliations:** 1Department of Applied Chemistry, Graduate School of Engineering, Osaka Prefecture University, 1-1 Gakuen-cho, Naka-ku, Sakai, Osaka 599-8531, Japan; 2Department of Applied Chemistry, Graduate School of Engineering, Osaka Metropolitan University, 1-1 Gakuen-cho, Naka-ku, Sakai, Osaka 599-8531, Japan

**Keywords:** thermoresponsive polymer, phase boundary, cleavage reaction

## Abstract

A new programed upper critical solution temperature-type thermoresponsive polymer was developed using water-soluble anionic polymer conjugates derived from polyallylamine and phthalic acid with cleavage-induced phase transition property. Intrinsic charge inversion from anion to cation of the polymer side chain is induced through a side chain cleavage reaction in acidic aqueous media. With the progress of side chain cleavage under fixed external conditions, the polymer conjugates express a thermoresponsive property, followed by shifting a phase boundary due to the change in polymer composition. When the phase transition boundary eventually reached the examined temperature, phase transition occurs under fixed external conditions. Such new insight obtained in this study opens up the new concept of time-programed stimuli-responsive polymer possessing a cleavage-induced phase transition.

## 1. Introduction

Controlling intermolecular interactions such as polymer-polymer and polymer-solvent interactions is of great importance for creating stimuli-responsive polymers [1,2,3,4]. For instance, thermoresponsive polymers can switch their physical properties such as soluble-insoluble (for linear polymer) [5,6], hydrophilic-hydrophobic surfaces (for two-dimensional surface: thin layer) [7,8,9], and swelling-shrinking (for a three-dimensional network: gels) [10,11] by regulating intermolecular polymer-polymer and polymer-solvent interactions at different temperatures [12]. Due to these unique properties, thermoresponsive polymers are used in chromatography [13], tissue engineering [14,15], and drug delivery systems [16,17]. Two types of thermoresponsive polymers have been reported, namely, lower critical solution temperature (LCST) [12] and upper critical solution temperature (UCST) [18]. Nonionic hydrophilic polymers such as poly (N-isopropylacrylamide) [19,20,21], poly (2-oxazoline) [22,23], poly (vinyl methyl ether) [24,25], and poly (oligoethyleneglycol methacrylate) [26] have strong polymer-water (solvent) interactions due to hydrogen-bonding formation, thus, they are water-soluble polymers. However, temperature rise weakens the intermolecular polymer-water interaction, resulting in a phase transition of these polymers from water-soluble to water-insoluble at elevated temperatures, known as an LCST-type phase transition. In UCST-type polymers, the inverse phase transition from water-insoluble to water-soluble occurs with rising temperature. The key for determining phase transition type for thermoresponsive polymers is how to switch their intermolecular interaction modes, i.e., polymer-polymer and polymer-water interactions. As introduced herein, the reported thermoresponsive polymers exhibit their phase transition in response to temperature change across the phase boundary for switching intermolecular interaction modes.

Herein, a new type of programed stimuli-responsive polymer expressing thermoresponsive property with cleavage-induced phase transition was developed, i.e., we found that conjugates of polyallylamine (PAA) and phthalic acid with cleavable amide linkages show a UCST-type thermoresponsive property when achieving a critical cleavage degree under fixed external conditions (Figure 1). As the side chain cleavage of the conjugate proceeds, the carboxylate anion of the polymer side chain transforms to ammonium cation, i.e., the polymer-polymer electrostatic interaction becomes stronger with increase in reaction time, causing a boundary shift of UCST-type polymer phase transition toward high temperature. Hence, the phase transition of this conjugate occurs under fixed external conditions when the phase transition boundary reaches the observed temperature.

## 2. Results and Discussion

The conjugate of PAA and phthalic acid (PAA-Pht) with acid-cleavable amide linkage was prepared through the reaction between the primary amine of PAA and phthalic anhydride in weak alkaline condition (pH 9.5), as reported in our previous work [27]. After 24 h of reaction, unreacted phthalic anhydride was removed by dialysis against weak alkaline water to purify the obtained PAA-Pht. From ^1^H-NMR spectra of the obtained PAA-Pht, the peaks derived from phthalic acid groups newly appeared. In addition, the methylene peaks of the original PAA (δ ~ 2.3) were shifted to δ ~ 3.2 after the reaction due to amide-bond formation, demonstrating that the reaction between PAA and phthalic anhydride successfully proceeded (Appendix A). After the reaction, the peaks derived from the original methylene groups in PAA completely disappeared, indicating that all primary amines reacted with phthalic anhydride under this condition, and carboxylated PAA (PAA-Pht) was successfully prepared.

In the cleavage reaction of the amide linkage in the side chain of carboxylated PAA, the carboxylic acid can react with the carbonyl carbon of the amide bond with five-membered ring formation, and the free phthalic anhydride will leave the polymer, resulting in the ammonium cation moiety being generated in the polymer side chain (Figure 1) [28,29]. The phthalic anhydride will be hydrolyzed to phthalic acid, therefore the cleavage reaction is irreversible. Based on this reaction mechanism, the cleavage reaction of the amide linkage is dependent on the protonation degree of the carboxylates. The apparent p*K*a value for carboxylic acids of PAA-Pht was estimated using an acid–base titration. Appendix A shows the pH vs. protonation degree curve prepared from the acid–base titration, and the apparent p*K*a value of carboxylates in the side chain of PAA-Pht was estimated to be approximately 5.9. The effect of the acid-cleavable property on the solubility of PAA-Pht was monitored in an acidic aqueous solution (pH 5.2) with 4.0 mg/mL of PAA-Pht at 60 °C. In this experiment, we found that the polymer aqueous solution suddenly became turbid after several thousand seconds (Figure 1a). To evaluate this phenomenon, the time course of transmittance at 500 nm of PAA-Pht aqueous solution (4.0 mg/mL) was monitored at pH 5.2 and 60 °C (Figure 1b). The transmittance of the PAA-Pht aqueous solution was stably maintained at approximately 100% in the initial stage of incubation (<3000 s), indicating that the polymer was dissolved in the aqueous solution. The transmittance of the PAA-Pht solution suddenly decreased at a certain reaction time (approximately 4000 s), indicating that the acid-treated PAA-Pht was precipitated from the aqueous solution. This decrease in the transmittance might be induced through the cleavage reaction of amide linkages in the side chains. The cleavage of the polymer side chain was confirmed using ^1^H-NMR, in which the cleavage reaction was stopped by adding NaOH aqueous solution when the transmittance reached 90% (defined as 90%T). From the 1H-NMR spectrum of the collected polymer, the methylene peak adjacent to ammonium cation was clearly observed, demonstrating that the amide bond of PAA-Pht was cleaved during the incubation process. Approximately 20% of the amide bond was cleaved at 90%T under this condition (Figure 1c). Therefore, the phase transition from water-soluble to water-insoluble may be triggered by the cleavage reaction of amide bonds in PAA-Pht.

Because the reaction rate of the cleavage reaction depends on both the pH and temperature of the solutions, pH as well as temperature might influence the phase transition behavior of PAA-Pht. The effect of pH on the phase transition behavior of PAA-Pht was evaluated by monitoring the transmittance under varying pH values at 60 °C (Figure 2a). Here, we define the critical reaction time as the time required for the transmittance to reach 90%T. On lowering the pH of the aqueous medium, the critical reaction time became shorter (Figure 2b). This is reasonable because the carboxylates in the PAA-Pht side chain are more protonated at lower pH conditions, in which protonation of the carboxylate group is a crucial factor for promoting the amide-bond-cleavage reaction, as shown in Figure 1. Furthermore, the cleavage degree of amide linkages of PAA-Pht was gradually decreased as the pH decreased; in other words, the polymer phase transition occurred with a lower cleavage degree of PAA-Pht under more acidic conditions (Figure 2c). 

In addition, the effect of temperature on the phase transition behavior of PAA-Pht was evaluated under varying temperatures at a fixed pH (Figure 2d). The decrease in transmittance derived from the phase separation of polymers occurred at shorter reaction times with increased temperatures. The critical reaction time has good linearity in the Arrhenius plot (Figure 2e), suggesting that the phase transition behavior is dominated by the re-action rate of acid-catalyzed amide-bond cleavages in the polymer side chain. In ad-dition, it was confirmed that the cleavage degree of PAA-Pht at 90%T was slightly dependent on temperature (Figure 2f).

From the series of experiments, it can be considered that the phase transition of PAA-Pht is caused by the side-chain cleavage, and the critical cleavage degree of the side-chain amide linkage was determined by the external conditions. Interestingly, the phase transition of PAA-Pht was not induced by external condition change (such as temperature and pH) but by achieving the critical composition through the side-chain cleavage, i.e., this phenomenon can be explained as a cleavage-induced phase transition. The cleavage of the amide linkage in the PAA-Pht side chain proceeded in acidic conditions, and the cleaved side chain is then changed to an ammonium cation. That is, PAA-Pht becomes a polyampholyte, partially cleaved PAA-Pht, through cleavage of the amide linkages. As the cleavage reaction proceeds, the number of carboxylate anions in the polymer side chain is gradually decreased whereas that of the ammonium cation is conversely increased, and then, partially cleaved PAA-Pht could induce the conformational transition through intra/intermolecular electrostatic interaction. To date, it has been reported that several polyampholytes exhibit temperature-responsive phase transitions [30,31,32,33]. We noted that the critical cleavage degree of PAA-Pht at pH 5.2 slightly increased with an increase in temperature (Figure 2f); in other words, more effective electrostatic interaction between ammonium cations and carboxylate anions of the polymer side chain with a further progress in cleavage reaction is required for the phase transition in higher temperature, suggesting that the partially cleaved PAA-Pht has a potential UCST-type phase transition.

The matrix of polymer cleavage degree at different pHs and temperatures was prepared by measuring all cleavage degrees under different pHs and temperatures (Figure 3a). From this figure, the phase boundary of polymers with a certain composition (that is, the cleavage degree) could be estimated, and the boundary of polymer phase transition was gradually shifted with increasing cleavage degree of the amide linkage to higher temperatures and higher pH values. It was supported that the partially cleaved PAA-Pht was a UCST-type polymer from the change in transmittance against stepwise change in temperature (Figure 3b). By increasing the temperature from 50 °C to 60 °C after reaching 90%T, the transmittance of the partially cleaved PAA-Pht aqueous solution increased, indicating that the phase-separated polymers were partially dissolved. Furthermore, the increase of transmittance was repeatedly observed by a temperature change from 60 °C to 70 °C. The stimuli-responsive polymer reported here shows unique UCST-type phase transition based on the intrinsic trigger (i.e., side-chain cleavage) under constant ex-ternal conditions. Thus, the PAA-Pht is concluded to be a novel UCST-type thermoresponsive polymer based on a cleavage-induced phase transition.

In conclusion, PAA conjugates with phthalic acid have been developed as a programed thermoresponsive polymer with cleavage-induced phase transition property, where the intra/intermolecular electrostatic interaction is turned on by intrinsic charge inversion from anion to cation of the polymer side chain by a cleavage reaction in aqueous media. The interesting point of this polymer is that the phase transition occurred under fixed external conditions. From the series of experiments, the critical cleavage rate of the side chain was determined by several parameters including temperature and pH, indicating the boundary shift of programed polymer phase transition occurred by regulating the external conditions. It is considered that this new type of programed stimuli responsiveness based on a cleavage-induced phase transition expands the concept of a stimuli-responsive polymer by adding a temporal axis

## 3. Materials and Methods

### 3.1. Materials

PAA aqueous solution (MW: 15,000) was purchased from Nittobo Medical Co. Ltd. (Tokyo, Japan). Sodium hydrogen carbonate (NaHCO_3_) was purchased from Wako Pure Chemical Industries (Osaka, Japan). Sodium chloride (NaCl), sodium carbonate (Na_2_CO_3_), sodium hydroxide (NaOH), and phthalic anhydride were purchased from Nacalai Tesque (Kyoto, Japan). The phthalic anhydride was used after grinding using a mortar. PAA-Pht was synthesized by the reported procedure [27]. In brief, PAA was dissolved in carbonate buffer (2.5 mg/mL), and the pH of the solution was adjusted to 9.5. After the addition of phthalic anhydride (3.0 eq. to amine groups of PAA) into the solution, the reaction performed for 24 h at room temperature. After purification by dialysis, the PAA-Pht was corrected by lyophilization. Hydrogen chloride (HCl) was purchased from Kishida Chemical, Co. Ltd. (Osaka, Japan). All chemicals were used without further purification. Spectra/Por^®^ Membrane (molecular weight cutoff: 1000 Da) purchased from Spectrum Chemical Mfg. Corp. (New Brunswick, NJ, USA) was used after immersion in pure water for 30 min. Deionized water was obtained from a Millipore Milli-Q purification system.

### 3.2. Acid-Base Titration

PAA-Pht (40 mg) was dissolved in 150 mM NaOH aqueous solution containing 150 mM NaCl (6 mL). The polymer solution was titrated with 150 mM HCl aqueous solution at 25 °C using an automatic titrator (AUT-701, DKK-TOA CO. Ltd., Tokyo, Japan). From the titration curve, apparent pKa and protonation degree of these polymers at each pH condition were estimated.

### 3.3. Transmittance Measurements

PAA-Pht was dissolved in 150 mM NaCl aqueous solution. The pH value of the polymer solution was adjusted to 5.2. The transmittance of the solution at 500 nm in various conditions was measured using UV-Vis spectrophotometer (V-550, JASCO Coop, Tokyo, Japan). Polymer concentration: 4.0 mg/mL; temperature: 50, 60, 70, and 80 °C; pH: 5.0, 5.2, 5.4, and 5.6. In the case of monitoring transmittance against stepwise change in temperature, PAA-Pht (4 mg/mL) was dissolved in 150 mM NaCl aqueous solution (pH 5.2, 1 mL). The transmittance of the solution at 500 nm was measured using UV-Vis spectrophotometer. In temperature jump experiments, the temperature was initially settled at 50 °C, and the transmittance of the polymer solution at 500 nm was measured using a UV-Vis spectrophotometer. Then, the temperature was raised to 60 °C when the transmittance was decreased to 90%. The same temperature change from 60 °C to 70 °C was also carried out sequentially.

### 3.4. Determination of Cleavage Degree of PAA-Pht

When the transmittance derived from PAA-Pht reached 90%, the cleavage reaction was stopped by the addition of 1.0 M NaOH aqueous solution (10 μL). The solution was then freeze-dried and yielded the dry polymer as a white powder. The cleavage of the polymer side chain was evaluated by ^1^H-NMR (400 MHz, ECX400, JEOL Ltd., Tokyo, Japan) after dissolving into D_2_O containing a small amount of NaOD. The cleavage degree of PAA-Pht was determined from the peak area ratio of the methylene peak adjacent to the original amide bond (a in Figure 1) and that adjacent to ammonium cation formed by the cleavage reaction (b in Figure 1) in ^1^H NMR.

## Data Availability

The data presented in this study are available on request from the corresponding author.

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
