# Peer review of "Programed Thermoresponsive Polymers with Cleavage-Induced Phase Transition"

_molecules, 2022, doi:10.3390/molecules27186082_

Round 1

Reviewer 1 Report

Overall, this is a well-written manuscript, however, before being published, a few aspects must be taken into account:

- Authors should speak in the introduction part about real and potential applications of this type of polymeric systems.

- Authors should look for more actual references related to the topic, and include them at the introduction partt. Moreover, the number of works mentioned within the manuscript and listed in bibliography is too short.

- Do authors study the reversibility of the cleavage-induced phase transition? If not, authors must include that part in the manuscript, or even mention something about this and the efficacy an p possibility to occur.

Reviewer 2 Report

Comments:

Atsushi Harada et al reported Programed Thermoresponsive Polymers with Cleavage-In- 2 duced Phase Transition. My reading suggests that this manuscript requires minor revision for publication in this journal.

1. In Fig. 1, the author says, "Approximately 20% of the amide 95 bond was cleared at 90%T under this condition (Fig. 1c). What is the rationale for the 20% cleavage?

2. The author says, "The reaction rate of the cleavage reaction depends on both the pH and temperature of the solutions”. By the way, why does the transmittance appear low from the start at pH 5 and 80°C in Fig. 2?

Reviewer 3 Report

Dear Editor, 

I looked at the manuscript. The Authors report the development of a water-soluble anionic polymer conjugates derived from polyallylamine and phthalic acid with cleavage-induced phase transition property. They studies its responsive properties using physicochemical characterization methods. I have three comments:

i. "cleavage-induced" needs explanation in view of the experimental process,

ii. the authors give only  ref. 22 (https://doi.org/10.1021/acs.macromol.2c00795)or the preparation of the PAA-Pht conjugate. They should also give the synthesis procedure in detail,

iii. The authors should give details about the differences of the ref. 22 and this work.
